# Enhancing Bayesian Approaches in the Cognitive and Neural Sciences via Complex Dynamical Systems Theory

**Luis H. Favela** [1,2,*] and **Mary Jean Amon** [3]

1    Department of Philosophy, University of Central Florida, Orlando, FL 32816, USA
2    Cognitive Sciences Program, University of Central Florida, Orlando, FL 32816, USA
3    School of Modeling, Simulation, and Training, University of Central Florida, Orlando, FL 32816, USA
\*    Correspondence: luis.favela@ucf.edu

**Abstract:** In the cognitive and neural sciences, Bayesianism refers to a collection of concepts and methods stemming from various implementations of Bayes' theorem, which is a formal way to calculate the conditional probability of a hypothesis being true based on prior expectations and updating priors in the face of errors. Bayes' theorem has been fruitfully applied to describe and explain a wide range of cognitive and neural phenomena (e.g., visual perception and neural population activity) and is at the core of various theories (e.g., predictive processing). Despite these successes, we claim that Bayesianism has two interrelated shortcomings: its calculations and models are predominantly linear and noise is assumed to be random and unstructured versus deterministic. We outline ways that Bayesianism can address those shortcomings: first, by making more central the nonlinearities characteristic of biological cognitive systems, and second, by treating noise not as random and unstructured dynamics, but as the kind of structured nonlinearities of complex dynamical systems (e.g., chaos and fractals). We provide bistable visual percepts as an example of a real-world phenomenon that demonstrates the fruitfulness of integrating complex dynamical systems theory in Bayesian treatments of perception. Doing so facilitates a Bayesianism that is more capable of explaining a number of currently out-of-reach natural phenomena on their own, biologically realistic terms.

**Keywords:** Bayesianism; noise; linearity; nonlinearity

"The elegant body of mathematical theory pertaining to linear systems . . . tends to dominate even moderately advanced University courses . . . . The mathematical intuition so developed ill equips the student to confront the bizarre behavior exhibited by the simplest of discrete nonlinear systems . . . . Yet such nonlinear systems are surely the rule, not the exception, outside the physical science. . . . Not only in research, but also in the everyday world" [1], p. 467.

## 1. Introduction

Perhaps the one thing cognitive scientists, neuroscientists, and philosophers of science can agree on is that there is no clear frontrunner for the title of "Grand Unified Theory" of brain structure and function. Contenders vying for the title include coordination dynamics [2], free-energy principle [3], network theory [4], neural Darwinism [5], and neural reuse [6], just to name a few. Although it is unlikely that any of those will be crowned as the champion any time soon, there are frontrunners for theories of particular aspects of the brain. An increasingly popular frontrunner in recent years is *Bayesianism* (e.g., [7], p. 16; [8], p. xi; [9], p. 1; [10], p. 1).

In the cognitive and neural sciences, "Bayesianism" refers to a collection of concepts and methods stemming from various implementations of Bayes' theorem, which is a formal way to calculate the conditional probability of a hypothesis—broadly construed—being true based on prior expectations and updating priors in the face of errors. Assorted iterations of Bayes'

theorem have been fruitfully applied to describe and explain a wide range of cognitive and neural phenomena, such as emotion and homeostatic regulation (e.g., [11,12]), neural population activity (e.g., [13–15]), and perception and action (e.g., [16–20]). In addition to data analytic and modeling work, such as those just mentioned, Bayes' theorem is also at the foundation of a variety of theories aimed at encompassing more general brain structures and functions. These include the theories of active inference (e.g., [21,22]), Bayesian brain (e.g., [3,8,23–26]), predictive coding (e.g., [22,27,28]), and predictive processing (e.g., [29–32]). In addition to these theories and their related data analyses and modeling work becoming increasingly popular, they also seem to have broad appeal. For example, proponents of both brain-centric—such as computational and representational (e.g., [9,33])—and non-brain-centric—such as embodied (e.g., [7,34])—approaches to cognition have appealed to Bayesianism to support their work. Bayesianism has even been applied to research on plant intelligence (e.g., [35]). The popularity of Bayesianism has led some to discuss it in terms of a "paradigm shift" in the cognitive and neural sciences (e.g., [36]).

While Bayesianism is popular, it is not without criticism (for review see [37]). Critical lenses have been drawn upon numerous aspects of Bayesianism including, but not limited to, the epistemic justification of its claims (e.g., [38]), the scope of its applicability (e.g., [39]), and whether it provides empirical tests of its actual theory (e.g., [40]). Here, we draw attention to a set of interrelated shortcomings of Bayesianism that we believe have been at a minimum underrated and at a maximum disregarded by its proponents in the cognitive and neural sciences. Specifically, the calculations and models typically employed in Bayesianism-based research are predominantly linear (i.e., as opposed to nonlinear) and they treat noise as random and unstructured (i.e., as opposed to deterministic and structured). As we will argue, the consequences of not addressing these issues is that the calculations and models produced by Bayesianism frameworks will not be biologically realistic. Consequently, while such work may provide models with accurate predictions, they will not *explain* neurophysiology or experimental data from the cognitive sciences.

In order for Bayesianism to be a viable resource for the scientific investigation of cognition and neural systems, it needs to be enhanced by features that facilitate analyses and models of targets of investigation in more biologically realistic terms. To do so, we argue that Bayesianism needs to take two steps: first, make more central the nonlinearities characteristic of biological cognitive systems, and second, treat noise not as random unstructured dynamics, but as the kinds of structured nonlinearities exhibited by complex dynamical systems, such as catastrophe flags, chaos, and fractals. If successfully integrated, complex dynamical systems theory will facilitate a Bayesianism that is more biologically realistic by emphasizing dynamics of the noisy nonlinear structured kind and not the linear unstructured sort that they already calculate and model.

In the next section, we provide a brief overview of Bayesianism, with subsections including an introduction to Bayesianism, an explanation of how linearity is inherent to traditional Bayesian analyses and models, as well as the typical treatment of noise in such approaches. After, we discuss what complex dynamical systems theory means in the current context, with emphases on nonlinearity and structured noise. After, we provide an example of the successful integration of complex dynamical systems with Bayesianism centering on bistable visual percepts. Lastly, we discuss challenges and future directions, followed by concluding remarks.

## 2. Bayesianism in the Cognitive and Neural Sciences

The current paper does not aim to provide a literature review of Bayesianism. Instead, we focus upon aspects that we take as common to its application in the cognitive and neural sciences and highlight those features that make clear the shortcomings we have identified and aim to address. Moreover, it is crucial that it is clear that the specific relationship between Bayesianism and its appearance in the cognitive and neural sciences can be quite varied. This is due in no small part to the fact that there are at least a few different kinds of relationships that can be identified. One is in terms of the kinds of

theoretical commitments held and how they inform explanations, such as the role of action and embodiment (e.g., [31]). Another is how to interpret Bayesian statistics, for example, as default, frequentist, or subjectivist (e.g., [41]). A third, and the most significant for our current purposes, is the way to interpret Bayesian modeling.

In terms of the third point, there is a stronger interpretation in which the brain is literally "Bayesian" [24,25]. That is to say, various features of the brain, such as computations and neuronal firing, are Bayesian in nature [26,29], p. 105. With respect to modeling, such an interpretation is *realist* in that "when a Bayesian model is explanatorily successful, we have reason to regard the model as approximately true" [9], p. 9. On the other hand, a weaker interpretation is the *instrumentalist* view, in which Bayesian methods are fruitful when analyzing and modeling behavioral, cognitive, and neural phenomena purported to involve forms of Bayesian perception (e.g., predictive processing; e.g., [8,42,43]). Although such models may provide successful predictions, such empirical successes do not give reason to then understand the target phenomenon as *being* Bayesian. For example, though a Bayesian model could accurately predict the results of an experimental task on visual perception (e.g., [21]), which would treat the behavior as a form of inference based on priors, that the model defines the phenomenon *as if* it has priors does not literally mean the visual system utilizes priors (cf. [9]). With that said, we now provide a brief explanation of Bayes' theorem and demonstrate the linearity at its core.

### 2.1. A Brief Introduction to Bayes' Theorem

As stated above, our aim here is not to provide another comprehensive introduction. We refer readers to the many excellent introductions to Bayesian analysis, modeling, and theory (e.g., [44–49]). To start, it is important to make clear that there is no single "Bayesianism" (e.g., there are default, frequentist, and subjectivist versions; [41,45]). Nevertheless, there are some core ideas common to its various forms, namely, *Bayes' theorem* [44] (see Equation (1)):

$$P(H|D) = \frac{P(D|H) \times P(H)}{P(D)} \tag{1}$$

Here, $H$ refers to a hypothesis and $D$ to data. $P(H)$ refers to the prior probability of belief in $H$ before data $D$ is obtained. $P(D)$ refers to the probability that the data $D$ reflects the actual state of affairs. Thus, $P(H|D)$ captures the probability of hypothesis $H$ being true after data $D$ is obtained. While Bayes' theorem can be understood in this straightforward way, it is important to recognize that depending on the version and manner of its implementation, there are a potentially wide range of deep issues concerning its mathematical structure, such as abstract versus concrete, finite versus countable additivity, and measure theory versus linear theory, just to name a few [44], pp. 161–164.

With respect to one of its roles in the cognitive sciences, Bayes' theorem underlies the claim that various cognitive and perceptual capacities are the result of a probabilistic computational process. A common example of perception in Bayesian terms is when somebody hears a sound and believes it is a song they know, but it turns out that the sound is not the song they thought it was or any kind of music altogether [30,50]. In such cases, when people realize the sound is not what they thought it was, the next time they hear it they do not make the same mistake of believing it to be a familiar song. This process of making predictions and updating future expectations based on current errors is described in terms of Bayes' theorem (or rule). When contributing to a model that attempts to explain a phenomenon as an instance of Bayesianism (e.g., predictive processing), Bayes' theorem dictates that the predicted structure of a set of data (e.g., sensory input) will be based on a set of candidate explanations that have prior credibility. When new data are presented, credibility is shifted towards the better candidate explanations and away from those that do not explain as well [48]. This conception of hypothesis testing is aligned with Bayes' theorem, or the "mathematical relation between the prior allocation of credibility and the posterior reallocation of credibility conditional on data" [47], p. 100. A concrete example will help unpack this.

Referring back to the example above of somebody hearing a sound and believing it to be a sound they know, here, we will use the song "White Christmas" by Bing Crosby (Figure 1) as an example. In this case, *H* refers to the hypothesis that the sound heard is the song "White Christmas", and *D* refers to data such as the sound. Thus, $P(H|D)$ captures the probability of hypothesis being true—"I believe I am hearing Bing Crosby's 'White Christmas'"—after the data (i.e., sound emitted from a car stereo speaker) are obtained. As it turns out, in this case the sound was not "White Christmas". It was Los del Rio's "Macarena Christmas", which sampled a part of "White Christmas". The next time you turn the radio on, since your prior hypothesis has been updated, when you hear "Macarena Christmas", you are correct to hypothesize that the sound could be *either* Bing Crosby or Los del Rio. The question now is, what makes Bayes' theorem, and models based on it, linear and why is that a problem?

**Figure 1.** Bayesian perception. Bayesianism postulates that perception is a process of prediction (i.e., internal models, or prior; see figure left side) from incoming sensory input (i.e., from an unpredicted environment; see figure right side). Prediction error is the difference between current predictions and sensory input. If the sensory input does not fit with the prediction (i.e., prior), then the internal model updates (i.e., future prediction, posterior, which becomes new prior). Here, the current prediction is the song "White Christmas" by Bing Crosby. However, the sensory input is actually Los del Rio's "Macarena Christmas", which becomes the future prediction. (Figure inspired by [11]. Parts of figure modified and reproduced with permission from FlyClipart. CC BY-NC 4.0.).

*2.2. Linearity and Its Consequences for Bayesian Modeling*

To understand the shortcomings of Bayesianism in the cognitive and neural sciences that we identify, it is essential to recognize that Bayesian models are typically limited to characterizing target phenomena as essentially linear. While *not all* applications of Bayesianism to cognitive and neural phenomena are constrained to linear models, those that are remain problematic for two major reasons (for examples and discussion, see below Section 2.2.2). First, those phenomena are predominantly nonlinear in nature. Second, much work that does attempt to account for the "nonlinearities" of said phenomena in Bayesian terms do so in ways that relegate nonlinearity to unstructured noise, which is contrary to the structured "noise" they commonly exhibit. Consequently, we encourage increased incorporation of nonlinear features of the structured kind in Bayesianism. However, first, what do we mean by "linearity" in Bayesian models?

As stated above, there are several forms of Bayesianism. In the same way, there are various kinds of linearity relevant to the study of cognitive and neural systems (e.g., static and ergodic). We aim here at a general sense of understanding linearity. At its most basic, "linearity" refers to a mathematical relationship in which outputs are proportional to inputs [51], p. 7 (see Equation (2)).

$$f(x + y) = f(x) + f(y) \tag{2}$$

Here, a function $f$ is defined as the sum of $x$ and $y$, which is equivalent to the sum of the functions $f(x)$ and $f(y)$ [52], p. 52. Another way to think about linearity in this way is to conceptualize $f$ as a system comprising the sum of parts $x$ and $y$, which is equivalent in some sense (e.g., weight, force output, etc.) to the sum of systems $f(x)$ and $f(y)$. In short, whether explained as a single function (or system) or two summed functions (or systems), $f(x + y)$ is linearly equivalent to $f(x)$ and $f(y)$.

Bayesian analyses and modeling can be considered linear when they overlap with core assumptions of frequentist statistics, in particular, a number of assumptions underlying both types of statistics. Target practice serves as a metaphor to help to understand these assumptions, especially in terms of naturally occurring variation during repeated trials [53,54]. While firing a gun at a shooting range, each bullet strike varies in how close it lands to the target's bullseye, with strikes clustering closer together around the center and becoming progressively more dispersed away from the bullseye (Figure 2). Even with the most experienced shooter, the bullet's trajectory will vary as a function of many influences, including the bullet's shape, amount of gunpowder, wind, etc. A bullet's deviation from the target can therefore arise from a variety of independent perturbations that have an additive effect on the overall trajectory. The metaphor of target practice underscores the basic logic of the *normal distribution* developed by Gauss [55,56] and Laplace [56,57]. Much like the concentration of bullet strikes around a target, Laplace reasoned that measurement error is often the result of many random, weak, and independent sources of variation that have an additive effect on measured behavior. The sum effect of these numerous perturbations is a bell-shaped or "Gaussian" distribution of measurements that are relatively symmetrical around a central tendency. This distribution and variability of events is known as the *central limit theorem*, which is a statistical theorem that assumes that the sum of numerous independent influences will have a normal or upside down 'U'-shaped distribution [58]. Normal distributions are also the result of a *frequentist* view of events, that is, each event (e.g., bullet fired at a target) is considered isolated such that each event outcome is equally probable (de Finetti in [44], p. 3). Consequently, events falling along normal distributions are conceived of as resulting from stationary processes. Processes, or events, are stationary when temporal ordering does not affect the probability of their distribution, such that mean and variance do not change with time [59].

The logic of the Gaussian distribution is rooted in systems where the probability of observed outcomes varies based largely on chance. Scholars traditionally framed variation during coin tosses, dice rolls, measurement accuracy of star positions, and target practice as being randomly influenced with no common thread of history linking observations [54]. As a result, linear statistics have been situated in a context that emphasizes *logical variation*. Logical variation is the difference between one discreet measurement and another, where variation is not influenced by history. This contrasts with *historical variation*, or fluctuations that occur as a process, unfolds over time [54]. Thus, while the Gaussian distribution has been the basis of numerous statistical advancements, its development left statisticians with the formidable task of applying principles derived from logical variation to natural systems whose behavior is influenced by history. While debates concerning the advantages and disadvantages of commitments to assumptions of logical and historical variation occurred, a number of linear statistics were developed, including correlation, simple regression, and multiple regression. All the while, the "perpetual flux" of natural systems was largely ignored as worthwhile targets of investigation in favor of evaluating discrete measurements [54]. Consequently, each of these statistics relied on the central limit theorem, carrying with them assumptions inherent to the Gaussian distribution.

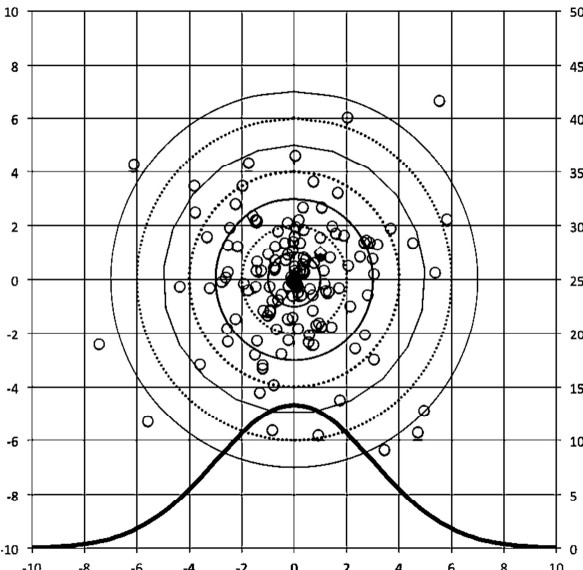

**Figure 2.** The normal distribution of bullet strikes on a target. Target practice shooting serves as an example of the assumptions of linear statistics, particularly the natural variation of a normal distribution purported to occur during repeated trials (figure—center). Normal distributions illustrate the central limit theorem, or the frequentist view of variability of events that assumes that the sum of numerous independent influences will have a normal or upside down 'U'-shaped distribution (figure—bottom).

Due to the fact that linear statistics account for logical variation in independent and dependent variables, additional fluctuations within biological or social systems (i.e., historical variation) are commonly relegated into the category of *unsystematic noise*. Unsystematic perturbations are assumed to be weak and occur randomly, such that their effects will cancel each other out. The variation accounted for by the primary variables of interest (i.e., independent and dependent variables) must exceed the variation produced by "unsystematic noise" by approximately two standard deviations in order for the relationship between these variables to be considered statistically significant using the null hypothesis significance testing. To the extent that the cognitive and neural sciences aim to explain neurophysiology and experimental data from the cognitive sciences, this holds true for applications of Bayesian data analyses as well. Accordingly, the ability to uncover statistically significant results rests on the idea that the noise within a system is unsystematic.

In this way, the general linear model and linear analysis served as the default organizing theory for the sciences of mind, especially psychology, and then the cognitive and neural sciences. That is, the bulk of the quantitative theories in behavioral and social sub-disciplines can be reframed as a version of a regression equation: factors, processes, latent variables, and other constructs can be present, absent, or combined in weighted sums to approximate empirical measurements. Consider stage-wise developmental theories, feature-based perceptual theories, additive factors theory, and the double dissociation and dual-process theories of psychology and cognitive science: they are all frequentist and stationary at their core.

All statistical procedures require nontrivial, a priori assumptions. In the case of linear statistics, they are situated within a context that has largely emphasized logical variation and ignored historical variation. The widespread generalization of the Gaussian distribution to biological systems (e.g., human action, perception, and cognition) has left many statistical procedures to rely on the following four underlying assumptions: First, *independent* perturbations have an *additive* effect on global behavior. Second, experimental measurement will result in a *Gaussian* distribution. Third, systems are best characterized by *logical variation* and not historical variation. Fourth, variation not accounted for by an experimental manipulation is weak, *random noise* that will cancel out over-repeated obser-

vations. Note that the third and fourth assumptions focus on the concept of stationarity. The message implicit in these assumptions is that the systems we study in the cognitive and neural sciences, and those that we characterize with linear statistics, are not influenced by history and—apart from the effect of the manipulation—are relatively isolated events that are static over time.

The concepts of linearity are so thoroughly entrenched in our statistics curricula (cf. [1]) that many have a difficult time understanding what is and is not a linear model, let alone why one would question the significant effects of thinking in only linear terms. Accordingly, let us take a moment to expand on Equation (2) presented earlier by adding in interaction ($\times$) and error ($\varepsilon$) terms (see Equation (3)):

$$f(x + y) = f(x) \times f(y) + \varepsilon \tag{3}$$

One might incorrectly assume that adding in an interaction term (e.g., $\times$) will shift the model from accounting for additive (qua linear) effects to multiplicative (qua nonlinear) effects. However, interaction terms such as those above only reveal linear changes in variable $x$ as a function of variable $y$. This is because linear models are a "statistics of averages" that collapse a dataset into mean values for each grouping variable. Readers can likely recall basic plots with significant interaction terms that have two lines crossing one another to form an 'X' shape to represent the linear interaction between two or more variables. In addition, one might assume that the random noise term ($\varepsilon$) accounts for "nuisance" nonlinearity in the data to uncover more valuable and interesting linear relationships. However, random noise is only one type of nonlinearity and happens to be one of the least interesting one could study, as we are generally interested in uncovering systematic relationships and rules (or laws, universality, etc.) within natural systems. To assume a random error term accounts for nonlinearity would mistakenly reduce the many well-established and systematic patterns of nonlinearity recorded across natural systems, which we will describe in greater detail below.

The basic assumptions of linearity (e.g., normality and homogeneity of variance) are rarely checked and are altogether violated, leading some researchers to describe common statistical procedures as "opportunistic" [60]. For the minority of researchers that do attempt to satisfy underlying assumptions of their linear analyses, they often rely on techniques such as averaging, outlier censorships, transformations, and the robustness of *t*- and *F*-tests. There are strategies to deal with a number of statistical violations common to linear statistics, but other assumptions of linearity are mismatched with the perpetually fluctuating systems often studied in the cognitive and neural sciences. It is rarely called into question that linear statistics treat changes in a system outside the effect of the independent variables as unsystematic noise that will cancel out over-repeated observations. Given that Bayesian modeling typically adheres to the above four assumptions of linear statistics, so too do they fall victim to the limitations facing other linear statistics. For clear depictions of this, let us examine Bayesianism in terms of a modeling approach for cognitive and neural phenomena.

### 2.2.1. Examples of Linearity in Bayesian Modeling

In terms of a modeling approach, Bayesianism typically adheres to many of the assumptions of frequentist statistics, including stationarity and linearity [44], p. 19. It is not difficult to substantiate this claim. In an overview of Bayesian models of perceptual organization, for example, it is stated that "Bayesians routinely assume a Gaussian (normal) prior distribution" [16], p. 1017. The following are a few additional examples that demonstrate such commitments: When utilizing Bayes' factor as a means to assess competing hypotheses, a normal distribution of data is usually assumed (e.g., [61]). Frequentist statistics are often needed to evaluate inferences made based on Bayesian analyses, that is, the conclusions of Bayesian analyses can require further validation by frequentist methods [62]. Additionally, before a Bayesian approach is applied to certain data sets (e.g., large sample

sizes), traditional frequentist statistical assumptions are often appealed to in order to make approximations about the data, for example, the assumption of normal distributions [62].

The need to supplement Bayesian methods with traditional statistics is particularly evident in the cognitive and neural sciences, for example, regarding perception where experimental work has usually considered simple tasks based on static variables (e.g., [63]) and underlying neuronal dynamics are treated as linear (e.g., [64]). Consequently, simple perceptual tasks based on static variables and linear neuronal dynamic models run the risk of being neither ecologically valid nor biologically realistic. This has been true of many popular neurobiologically inspired models as well, including connectionism. Although it is easy to find research on connectionist models as early as 1990 that include "nonlinear" parameters, those tend to fall victim to the same linear limitations discussed here. For example, the nonlinearities that are produced by various parameters in connectionist networks are limited to producing backpropagation and feedback, as well as randomness in forms such as stochasticity [65]. However, such parameters do not produce the kinds of interesting nonlinearities common to biological systems, such as those exhibiting structured noise discussed below (see Section 3). Before moving on to the treatment of noise in Bayesianism, it is helpful to provide a concrete and intuitive example of the way linearity informs Bayesianism, namely, the case of visual perception.

In general, Bayesianism treats visual perception as a form of inference (e.g., predictive processing; [29,32,43,66]. The inferences are modeled in the form of Bayes' theorem, which center on the probability of a hypothesis $H$ being supported by the probability of the data $D$ supporting that hypothesis being true, $P(H|D)$. In this way, visual perception can be modeled as follows (Equation (4)):

$$P(state|sensory\ input) = \frac{P(sensory\ input|state) \times P(state)}{P(sensory\ input)} \tag{4}$$

Here, the hypothesis ($H$) is the state of the world (*state*) that the perceiving system expects and the data ($D$) is the information that system receives. Thus, in accordance with Bayes' theorem, $P(sensory\ input|state)$ is the *likelihood* (i.e., probability) that the system's prior assumption about the state of the world ($P(state)$) reflects the information received ($P(sensory\ input)$), which results in the posterior belief ($P(state|sensory\ input)$). In this model, the posterior state of the system results from multiplying the prior (($P(state)$)) by the likelihood ($P(sensory\ input|state)$) and normalizing via summing of all possible *sensory input* and *state* values to one [8], p. 6. Hence, if the sensory input does not fit with the prior assumption, then the posterior will be normalized and result in new prior states. While much of the details of the computations have been left out here for the sake of brevity, it may remain the case that it does not seem intuitive how this model captures visual perception. To do so, consider the following example.

$P(state)$ could be the hypothesis that the image presented to me, ($P(sensory\ input)$), is a random collection of black shapes on a white background (Figure 3a). Thus, $P(state|sensory\ input)$ is the probability of *state* being supported by *sensory input*. Now, consider if the sensory information changes, and instead of a seemingly random spread of black shapes on a white background, the image has a line drawn around part of it (Figure 3b). Whereas your prior belief was that there is nothing of substance in the image, the added information (i.e., new line) reveals a Dalmatian dog sniffing the ground. Consequently, your perceptual system has been updated with a posterior belief. From that moment forward, when you see that same image, with or without the line, your new prior belief is to see a Dalmatian dog sniffing the ground (the Dalmatian dog illusion has been utilized as an example for work on Bayesian neural coding; e.g., [8]).

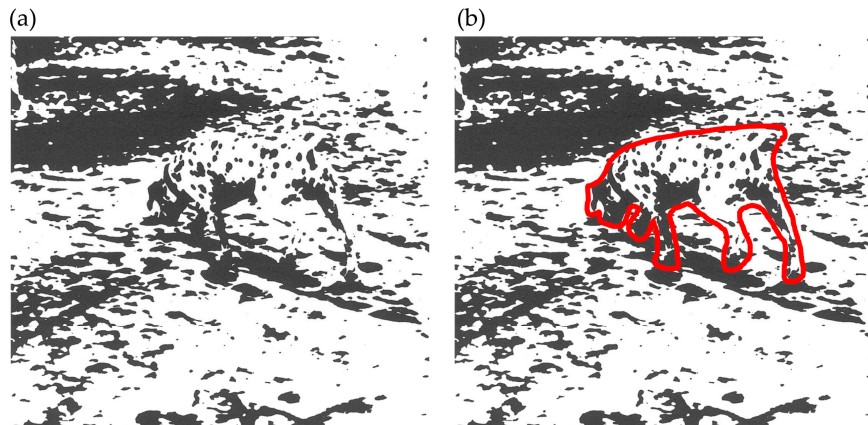

**Figure 3.** Hidden figure. (**a**) An apparently random collection of black shapes on a white background. (**b**) When a portion of the black shapes are outlined, it becomes evident that there is a Dalmatian dog sniffing the ground in the image. (Images based on [67]).

It is worth emphasizing here the importance of experience and expectation in accounting for the experience of this illusion. One's ability to eventually perceive the Dalmatian dog among the seemingly random collection of black shapes is likely in large part due to having seen such figures before (i.e., experience) and having a sense of the range of potential shapes such a figure can confirm to (i.e., expectation). These two points are illustrated by the hollow mask illusion. Here, under particular conditions, a three-dimensional shape with a concave inner surface will appear as if it is a concave outer surface. For example, if one is looking at a mask of a human face when turned away—that is, when they are looking at the concave side—it is common to view the mask as facing towards you, with such features as a nose that extends outward. This perception may be accounted for by the fact that the perceiver has experiences and expectations of human faces to look a certain way, namely, to be convex. However, research has suggested that those who do not experience face perception in neurotypical ways, such as autistic or schizophrenic subjects, are not as sensitive to the illusion [7,68]. Accordingly, Bayesian modeling and understanding of such phenomena ought to take into account such states of an individual when calculating prior probabilities.

This example is straightforwardly captured by Bayes' theorem and is conceptualized as a Bayes linear approach [69]. The Bayes linear approach is applied to problems where we want to combine prior judgments of uncertainty (*H*; e.g., belief that the image is a random collection of black shapes) with observational data (*D*; e.g., seeing apparently randomly placed black shapes on a white background; Figure 3a), and we use expected value (i.e., the image *really* does not have anything of substance to it) to express the perceptual judgment instead of probabilities [69], p. 1. When presented with the second image (Figure 3b), the new observational data (*D*; e.g., seeing lines drawn around some of the black shapes on a white background) does not match our prior judgment (*H*; e.g., the belief that the image is a random collection of black shapes), which results in a new expected value (i.e., the image has substance, namely, a Dalmatian dog sniffing the ground).

In this way, the Bayesian model of visual perception further demonstrates the four assumptions of linear statistics [69], p. 2: First, the independent perturbations (i.e., sensory data) are additive in that they are treated as uncorrelated sums. Second, the distributions of the calculations will be Gaussian, which the calculations require during normalizing in order to achieve a value of one (cf. [8,18]). Third, variation is accounted for in terms of other measurements and not previous states. Fourth, variation in beliefs (i.e., hypotheses and priors) are reduced or canceled out by variation in the data, as is variation in data reduced or canceled by variation in said beliefs ([69] makes explicit reference to similarities with de Finetti [44], p. 3). If we are correct, and if linearity is at the heart of Bayesianism, then such approaches in the cognitive and neural sciences will carry an additional assumption,

namely, a view of noise as random and unstructured features of targets of investigation. In the next section, we discuss noise and argue that prior attempts to incorporate it into Bayesian approaches to modeling cognitive and neural systems fall short.

2.2.2. Examples of Noise in Bayesian Modeling

In our discussion thus far, we have noted that there are multiple forms of Bayesianism (e.g., default, frequentist, and subjectivist; [41]). Still, as we have argued, linearity is at the heart of the Bayesianism commonly employed in the cognitive and neural sciences. What is more, nonlinearity is scarcely mentioned. For example, Ma, Kording, and Goldreich's [19] textbook on Bayesian models of perception and action mentions 'nonlinear' ('nonlinearly,' etc.) only a few times and predominantly in terms of errors and randomness (e.g., Section 6.6, B.12.5). Our explication of Bayes' theorem as demonstrating frequentist and stationary views underlies the assessment of Bayesianism as centrally being a type of linear analysis in line with linearity is a treatment of "noise" in terms of unsystematic perturbations that are assumed to be weak and occur randomly, such that their effects will cancel each other out (Section 2.2). With that said, work has been conducted to attempt to incorporate nonlinearity into Bayesianism, for example, Bayesian model comparison of neural activity [13,70] and natural image reconstruction [14]. One attempt to incorporate nonlinearity in a Bayesian approach is by way of *predictive coding*, which is an attempt to model the properties of neural networks underlying perception: "Predictive coding postulates that neural networks learn the statistical regularities inherent in the natural world and reduce redundancy by removing the predictable components of the input, transmitting only what is not predictable (the residual errors in prediction)" [27], p. 580. This postulate is formalized via Bayes' theorem [27], p. 581 (Equation (5)):

$$P(\boldsymbol{r}|\boldsymbol{I}) = P(\boldsymbol{I}|\boldsymbol{r})P(\boldsymbol{r})/P(\boldsymbol{I}) \tag{5}$$

$P(\boldsymbol{r}|\boldsymbol{I})$ is the model of the visual system, for example, whereby an internal representation of the external world is predicted based on sensory data. Here $\boldsymbol{I}$ is the probability of an image given hidden internal model parameters $\boldsymbol{r}$ (e.g., underlying neural network activity). Thus, for visual input $\boldsymbol{I}$, the system selects parameters $\boldsymbol{r}$ to maximize the posterior probability that the system's representation of the world is accurate. This occurs via a straightforwardly linear process [27], pp. 581–583, 585, 588, 590. While predictive coding models are standardly linear (for review see [71]), there have been attempts to supplement them with nonlinear features.

While Friston has developed related Bayesian models that are only linear (e.g., [22]), his work with other colleagues has attempted to incorporate nonlinearity into their models of predictive coding (e.g., [3,72]). Friston [3] provides the following generative model of predictive coding (note that explaining all of the variables is unnecessary for our current purposes) in Equations (6)–(8):

$$p(\tilde{s}|\vartheta) = p(\tilde{s}|\vartheta)p(\vartheta) \tag{6}$$

where

$$s = g\left(x^{(1)}, v^{(1)}, \theta^{(1)}\right) + z^{(1)} \tag{7}$$

$$\dot{x}^{(1)} = f\left(x^{(1)}, v^{(1)}, \theta^{(1)}\right) + w^{(1)} \tag{8}$$

Here, $g^{(i)}$ and $f^{(i)}$ refer to nonlinear functions with causal and hidden states with parameters $\theta^{(i)}$ [3], p. 130. Although Friston does not specify values for the variables, he states that $z$ and $w$ contribute "random fluctuations" and "noise" [3], p. 130, which is intended to capture nonlinear characteristics. In other work, Friston and colleagues again describe such nonlinear functions as "hidden states" that contribute "observation noise" and "random fluctuations", among other characteristics [72], p. 1211. If we take these models as indica-

tive of current thinking regarding the relationship between Bayesianism and nonlinearity, then we must conclude that Bayesianism has yet to properly incorporate nonlinearity.

By treating nonlinear variables as random fluctuations and noise, Friston and colleagues are revealing that their Bayesian modeling remains committed to the assumptions of linear statistics. In particular, they demonstrate the fourth underlying assumption of linear statistics: variation that is not accounted for by an experimental manipulation is assumed to be unsystematic noise that is not influenced by historical variation. While Friston and colleagues incorporate "nonlinear variables", their status as random fluctuations and noise relegate their contributions as carrying no significant weight in a Bayesian model that remains committed to the assumptions of linear statistics—take notice of the fact that this work is explicitly committed to Gaussian distributions of data involved in predictive coding (e.g., [3], p. 130; [72], p. 1212).

Additional examples demonstrate how attempts at accounting for nonlinearity have been limited to notions of randomness and unstructured noise. First, Colombo describes two Monte Carlo algorithms as forms of Bayesian algorithms that attempt to model category learning, and are purported to be "non-linear, non-Gaussian" [23], p. 4828. However, this example's Monte Carlo algorithms are "nonlinear" in the sense that they deal with random numbers [73]. Second, Eliasmith and Anderson state that their approach to neurobiologically plausible models can incorporate Bayesian inference, with emphases on the representation of random variables in their models and the role of probability [74], p. 276. Again, such an approach fails to incorporate Bayesian modeling and nonlinearity in any sense other than randomness and probability. As we see, time and time again, when phenomena are described via Bayesian modeling, "nonlinearity" is equated with rudimentary characteristics such as multiplicativity, feedback, or just unstructured noise and randomness. Additionally, appeals to probability alone do not work as probabilities need not be nonlinear, such as linear probability models [75].

Other researchers have drawn attention to the challenges of integrating Bayesianism and nonlinearity. In work on populations of neuronal activity from a Bayesian approach, Ma and colleagues point out that restricting neuronal models to Bayesian methods will be limited because Bayesian inference is additive (i.e., linear) and real neurons are nonlinear [76], p. 1436. Similarly, in a review of Bayesian network analysis of functional magnetic resonance imaging (fMRI), Mumford and Ramsey emphasize that "most [Bayesian network] approaches assume linearity and Gaussianity" [77], p. 576.

Notwithstanding such aforementioned challenges with integrating Bayesianism and nonlinearity, it has been argued that currently popular forms of Bayesianism are not applicable to real cognitive and neural systems as they can only be applied to linear systems. Aguilera and colleagues [78] motivate this conclusion based on a number of points centering on the free-energy principle (FEP). The FEP asserts that any self-organizing, dynamical system existing at equilibrium with its environment minimizes free energy—or leans towards order—by performing forms of Bayesian inference [3,78]. In this way, FEP is theoretically applicable to any cognitive and neural system. In order to do so, Aguilera and colleagues [78] note critical assumptions required for FEP to be applicable: target phenomena must be weakly coupled non-equilibrium linear stochastic systems that meet a Markov blanket condition and inflexible restrictions on their solenoidal flows [78], p. 25. As they argue, for these assumptions to be valid, they can only hold for a narrow range of parameters that are only applicable to linear (stochastic) systems. While there is an increasingly large annual output of research purporting to apply Bayesianism in the cognitive and neural sciences alone, the requirements that Aguilera et al. [78] draw attention to for the FEP to be viable have not been directly applicable in any real biological system.

Bayesianism suffers from the critical overarching limitation of being inapplicable to many real cognitive and neural systems. This is primarily due to limitations stemming from commitments to linearity and a related narrow sense of the kinds of "noise" exhibited by such systems. In the next section we provide a way to understand nonlinearity and noise that, if properly integrated, can enhance Bayesianism in ways that facilitate applicability to

real cognitive and neural systems. Specifically, Bayesianism needs to be integrated with complex dynamical systems theory, which, as discussed in the next section, lay emphasis on the view of cognitive and neural systems as being complex systems, whose principal features are structured activity and organization that emerge from nonlinear dynamics, such as chaos and fractals.

### 3. Complex Dynamical Systems Theory

The structure and processes of truly linear systems are rare in nature and tend to be limited to artifacts (e.g., combustion engines, dishwashers, laptops, etc.). Despite that fact, the cognitive and neural sciences have primarily focused on conceptions of targets of inquiry as linear and have utilized linear methodologies. From Donders' development of a subtractive method to attempt to decompose response time performance into components [79], to the general linear model central to analyses of fMRI [80], linearity has been the rule and not the exception in scientific investigations of cognitive and neural phenomena. Theoretically, the issue is whether or not the assumption of linear spatial and temporal structures in cognitive activities and neural physiology is justified or not.

Nonlinearity is the rule and not the exception across the physical and social world [1]. At its most basic, "nonlinearity" refers to relationships among variables that are not linear. Rather than adopting frequentist assumptions of additive relationships and independence of components, nonlinear phenomena may be composed of interdependent components that exhibit exponential or multiplicative dynamics [51], p. 6. Across various kinds of literature, including the cognitive and neural sciences, it is common to understand phenomena as being "nonlinear" when they exhibit characteristics such as multiplicativity (see Section 2.2.2.) or random noise (see term $\varepsilon$ in Equation (3)), such as irregularity, stochasticity, etc. However, multiplicativity and random noise describe only the most rudimentary forms of nonlinearity. Examples of more "complex" and systematic forms of nonlinearity include circular causality, radical qualitative shifts in behavior or organization, various forms of structured noise (e.g., fractal scaling) or fluctuations over time, as well as chaos and self-organization. Such qualities are definitive of *complex dynamical systems*. In a review of nonlinear dynamical analyses, Stam [81] makes a similar characterization of neural dynamics that exhibits what we are here referring to as "complex dynamical systems". Stam draws attention to "a much more interesting repertoire of dynamics" [81], p. 2269, such as those exhibiting complexity, deterministic chaos, and fractal geometry.

There are ever more empirical data to support the claim that cognitive and neural systems are nonlinear systems. Furthermore, cognitive and neural phenomena are nonlinear in profoundly complex dynamical ways. Here is but a small sample: In regard to spatial organization, there are ample data to suggest that across micro- (e.g., molecular networks), meso- (e.g., synaptic clusters), and macro-scales (e.g., neuronal networks), the brain is fractal and multifractal in structure [82]. In regard to temporal characteristics, there is much research indicating nonlinear dynamics such as bifurcations and self-organization in axonal and dendritic plasticity [83], single-neuron activity [84], neocortical circuits [85], and neuronal networks [86]. At larger scales of behavioral and cognitive activity, there are data to suggest complex dynamics as well, for example, decision making [87], postural control [88], speech categorization [89], temporal estimation [90], and visual search [91], just to name a few.

Notably, the analysis of characteristics of complex dynamical systems such as those described above have been facilitated by a wide range of nonlinear analytic tools [92]. Although these categories have much overlap, for the sake of simplicity, we distinguish two groups of nonlinear methods: "modeling" and "time-series analysis" [93,94]. Typical modeling techniques when investigating nonlinear systems include agent-based (e.g., cellular automata), computational (e.g., simulations), dynamical (e.g., differential equations), and network (e.g., neuronal networks). Time-series analyses utilized when investigating nonlinear systems include dynamical correlation, entropy (e.g., sample entropy), fractal and multifractal (e.g., detrended fluctuation analysis), phase space reconstruction (e.g.,

attractor reconstruction, e.g., Lorenz attractor), recurrence quantification analysis (e.g., cross-recurrence quantification analysis), and wavelet (e.g., wavelet cross-spectrum).

A number of lessons can be taken from the theory and methods that comprise complex dynamical systems theory. The presence of complex dynamics in many natural systems—including those investigated via the cognitive and neural sciences—challenges the basic assumptions of linear statistics, including additivity (i.e., linearity), independence of perturbations (i.e., modularity), and logical, unsystematic variability. In contrast, the ubiquity of complex dynamics in natural systems suggests the following are more appropriate assumptions: First, nonlinear methods take into account cases when multiplicative feedback effects on global behavior are due to *interdependent perturbations*. Second, complex dynamical systems theory acknowledges that variation within a system is often *systematic* and reflects underlying processes that interact to support behavior. Third, natural systems are in a state of perpetual flux and are often best characterized by *historical variation*. Fourth, experimental measurement of natural systems that entails discrete, cross-sectional measurements will inevitably capture changes in a system that are due to complex, interaction-dominant dynamics (i.e., and not component-dominant dynamics).

The fifth lesson is especially important: Investigative frameworks based on complex dynamical systems theory have three complementary and interrelated features that drive experimental design, inform hypotheses, and facilitate explanation and understanding: qualitative features, quantitative features, and theoretical explanations or generalizable principles of how the biological system works [92,95]. For example, when attempting to provide an explanation and understanding of phenomenon *X*, dynamical methods such as differential equations could be used to model the temporal evolution of the system as well as plot the behavior in a phase space reconstruction. In this way, a quantitative and qualitative account of *X* is given. In order to *understand* the complex dynamic features of the system and to *explain why* it has those features, the quantitative and qualitative methods can be supplemented with an appeal to "principles", or rules that underlie the spatial and temporal structure of many complex dynamic systems.

Catastrophe flags and universality classes provide sources of generalizable principles appealed to in complex dynamical systems investigative frameworks. *Catastrophe flags* are a set of dynamical patterns that anticipate or accompany phase transitions (i.e., switching between states) in complex dynamical systems [96–98]. Hysteresis, multistability, and sudden change/jumps are three of the eight known catastrophe flags [96]. Observing hysteresis, or one of the other flags, strongly suggests the system will undergo a phase transition, which is a common feature of complex dynamic systems. The cusp model is a classic example of the application of catastrophe theory, which illustrates the quantitative and qualitative features when applying the associated principle [99]. If a dataset adheres to the cusp model, then it will exhibit a number of catastrophe flags, which further supports understanding that system as exhibiting complex dynamics (Figure 4). The cusp model has been successfully utilized in nonlinear dynamical modeling of various phenomena across the cognitive and neural sciences (e.g., [84,92,100,101]). For example, nonlinear cusp models have better explained human attitudes and behaviors in a number of research areas than linear models alone [102–104]. Moreover, preliminary work has integrated the nonlinear features of the cusp model with traditionally linear statistical approaches like the linear regression model [105].

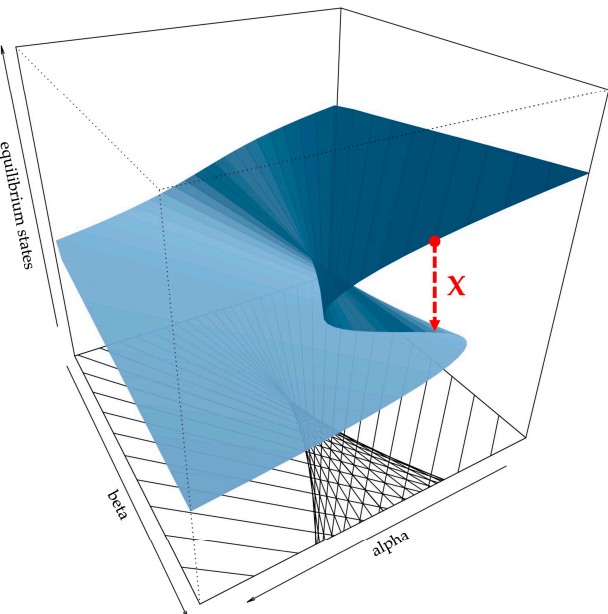

**Figure 4.** Cusp catastrophe model. The cusp model is a common application of catastrophe theory. It exhibits catastrophe flags such as hysteresis and bifurcation behaviors such as sudden jumps (X). A sudden jump is a phase transition, such as the sudden change in water in a teapot transitioning from liquid to gas. Here, *alpha* and *beta* could be temperature and pressure, with *equilibrium states* being the different conditions the system is attracted to and settles in (i.e., liquid, gas, and solid). Thus, at particular temperatures and pressures, water will jump (X) from liquid to gas. Successful applications of the cusp model include anxiety, beam elasticity, dog aggression, and stock prices [99]. (Modified and reproduced with permission from [106]. CC BY 3.0.).

*Universality classes* are another source of generalizable principles appealed to in complex dynamical systems investigative frameworks. They are distinct system behaviors that are determined by a few characteristics, take place across multiple spatial and temporal scales, and are substrate neutral [107,108]. Self-organized criticality (SOC) is one universality class that is observed across a variety of natural systems [109], including various brain/mind systems [110–113]. SOC is posited as an explanation for the ability of some nonlinear systems to exhibit multiple spatial and temporal scales that organize near critical states, which is a balance between two qualitatively different behaviors. Some argue that SOC explains why nonlinear systems are often dictated by power law features such as fractal self-similarity (e.g., [114]). Accordingly, if a system exhibits characteristics such as fractal structured behavior, then it may be appropriate to use nonlinear methods and provide explanations via principles derived from, for example, catastrophe theory and universality classes. Thus, if systems that have exhibited complex dynamics in other research are only assessed via standard Bayesian modeling approaches, then it is quite likely that key features of those systems are being overlooked.

The application of Bayesian paradigms has undoubtedly led to many successes in the cognitive and neural sciences. However, there remains a fundamental contradiction between the common statistical assumptions of additivity, linearity, and straightforward cause–effect relationships that underlie traditional Bayesian approaches and the complex dynamics inherent to the systems such assumptions are applied to. Here, we provided a very small sample of the large amount of literature that points to cognitive and neural phenomena as inherently complex dynamic systems, as well as the various methods used to properly characterize such dynamics. In doing so, we have attempted to motivate the point that nonlinearity is the rule and not the exception (cf. [1]), especially in regard to phenomena in the cognitive and neural sciences. It is important to make clear that such claims do not entail that all linear conceptions of phenomena and analyses ought to be cast

aside. Such treatments are appropriate for many investigative and explanatory purposes. Given the received view of linearity in the cognitive and neural sciences, it is worthwhile to make a strong case for the inclusion and often more appropriate role for nonlinearity as well. In the next section, we present an example that demonstrates the necessity of incorporating complex dynamical systems theory in a target phenomenon commonly investigated via Bayesianism.

## 4. Complex Dynamical Bayesianism

In order for the cognitive and neural sciences to properly integrate Bayesianism and complex dynamical systems theory, we recommend the following: First, models of target phenomena must incorporate variables that are nonlinear in the complex dynamic sense and not merely random or unsystematic noise. For example, parameters could be set to experimentally validated complex dynamic structures, such as fractals (i.e., $1/f$ signals). Second, if appropriate for a particular model, then nonlinear functions ought to be exponential or multiplicative and not just additive, as is the case for $g^{(i)}$ and $f^{(i)}$ above (Equations (7) and (8)). Third, models ought to be capable of exhibiting qualitative shifts in behavior or phase transitions and, thereby, exhibit catastrophe flags such as hysteresis and multistability. Fourth, principles such as those derived from universality classes ought to be able to contribute to prior probabilities (i.e., $P(H)$). For example, prior probabilities can be set to be in line with parameters that facilitate critical state behavior (e.g., SOC), such as those found across spatial and temporal scales from underlying neuronal dynamics (e.g., [85]) to gross-level behavior (e.g., [115]). As a consequence of adhering to these four recommendations, a *complex dynamical Bayesianism* will stress the roles that temporal structures play in understanding complex dynamic phenomena. Accordingly, its investigative frameworks will shift to explanations that leverage quantitative (e.g., differential equations) and qualitative (e.g., phase space reconstruction) tools, and appeals to principles such as those found in catastrophe flags and universality classes.

We offer bistable visual percepts as an example of a real-world phenomenon that puts our recommendations to work and demonstrates the fruitfulness of integrating complex dynamical systems theory in Bayesian accounts of perception in the cognitive sciences. A bistable percept is one that presents two or more stable states. Formally speaking, a dynamical system $\dot{x} = f(x)$ is bistable (or multistable) when it possesses two or more stable equilibrium points [116]. Examples of bistable visual percepts include the Rubin vase, Schroeder stairs, and—the philosopher's favorite—duck–rabbit (Figure 5a–c). Here, we focus on the Necker cube (Figure 5d).

If Bayesianism is correct in its current form, namely, with a linear foundation, then the transition between bistable states ought to occur at regular intervals and, thus, exhibit logical variation based on additive effects. Note that this point is not a consequence of the probabilistic nature of mathematic modeling of the phenomenon, but of the dynamics exhibited by the biological system. This is evident when, in our phenomenological experience, it does not seem true that transitions in how the Necker cube's orientation is perceived—i.e., from the front facing down and left to the front facing up and right, or vice versa—occur at regular intervals. Instead, the state transitions seem to occur with spontaneous irregularity, for example, requiring more or less time and effort to change orientations. In addition, it also seems that the more experience we have viewing the Necker cube, the more variation there is in the occurrence of those transitions. Since Bayesianism does not account for the irregularity and historically based variation of such perceptual experiences, then it must either be the case that our sense of timing regarding the state transitions of bistable visual percepts is inaccurate (a very real possibility; cf. [117]) or Bayesianism is inadequate to account for those features. Although evidence based on introspection can be methodologically problematic, since Bayesianism is purported to account for phenomenology as well (see Section 2.1 and Section 2.2.1), we believe that the Necker cube and other bistable visual percepts are cases that exhibit how the standard adherence to linearity and treatment of noise result in Bayesianism falling short.

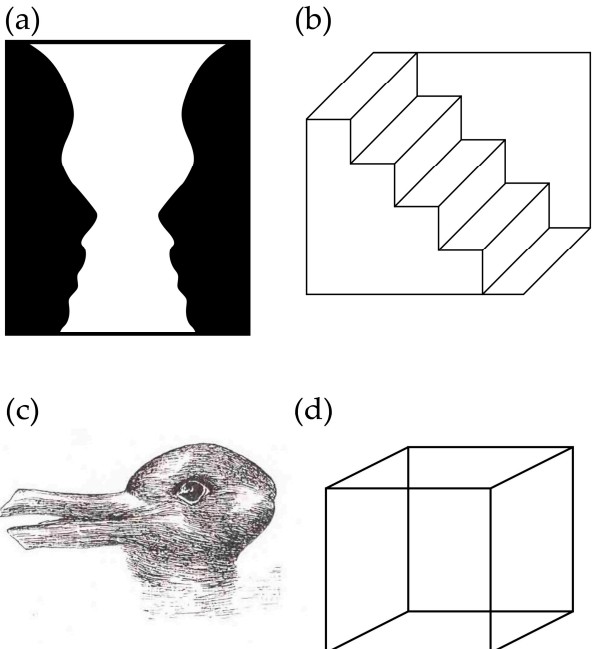

**Figure 5.** Examples of bistable visual percepts. (**a**) Rubin vase; (**b**) Schroeder stairs; (**c**) duck–rabbit; and (**d**) Necker cube. (Reproduced with permission from (**a**) Wikipedia, CC BY-SA 3.0; (**b**) public domain; (**c**) public domain; (**d**) Wikipedia, CC BY-SA 3.0.).

Two important points must be addressed before presenting our alternative. First, it is reasonable at this point to revisit the case of the Dalmatian image (Section 2.2.1; Figure 3). Did we not present that percept as an example to illustrate the linear commitments of Bayesian models of visual perception? Moreover, was the Dalmatian example not in the same class of bistable visual percepts as the duck–rabbit, Necker cube, and the like? The answers to these questions are yes and no, respectively. Yes, the Dalmatian image was presented as an example of linear Bayesian models. However, that image is not in the same class of visual percepts as the Necker cube. Consider your experience of the image (Figure 3a): Once you become aware of the substantive features of the image—namely, a Dalmatian dog sniffing the ground—you cannot go back and *unsee* it as such. This is because the posterior belief that there is a Dalmatian dog among the black shapes on the white background has become your new prior. Bistable visual percepts do not function as such, that is, your experience of the percept switches between the two states (either controlled or spontaneously) even after you have experienced it and your priors have been updated. The second point is to acknowledge that there is research applying standard Bayesian models to bistable visual percepts. We do not think this work demonstrates that standard, linear Bayesianism is appropriate to model bistable visual percepts in biological systems, such as mammals. The reason is that in order to make those models work, the original percept (e.g., "flanking" the bistable Necker cube with unambiguous cubes [118]), the participant (e.g., modulating neck posture [119]), or the task (e.g., changing the background and illumination of the environment [17]) had to be manipulated in order to produce the desired effect, which, in this case, is an unnatural visual response.

In order for a Bayesian model to account for the phenomenon of bistable visual percepts while maintaining both biological realism and ecological validity, it must integrate complex dynamical systems theory. As we discussed above, nonlinear methods account for, among other things, features such as historical variation. Hysteresis is one characteristic of some nonlinear phenomena that exhibit historical variation. Although there are various kinds of hysteresis, the basic idea is that a system's current state is dependent on its history [101]. Ferromagnetic materials provide a clear example of the history dependence resulting from the hysteresis effect. Once a material, such as iron, becomes magnetized, if it

is demagnetized, then it will take a different magnetic field to magnetize it again. Thus, the current state of the system depends on its previous state such that magnetization does not occur at context-free values. Ta'eed and colleagues [120] investigated hysteresis in the perception of the Necker cube. They demonstrated that nonlinear dynamical models of bistable perception were superior to linear models, primarily because the latter do not effectively account for multivalued functional relationships such as those involved in phase transitions during visual perception. Specifically, Ta'eed et al. demonstrated that bias (i.e., history), among other variables, affected the perception of the Necker cube upon subsequent presentations, thereby indicating a complex dynamic process. Along those lines, Haken [83,121] has also conducted experimental work on hysteresis exhibited during visual perception of bistable percepts. Although some kinds of visual perception may be amenable to explanatory accounts based on linear assumptions (e.g., [122]), the work of Ta'eed and colleagues demonstrates that others require integrating nonlinear dynamics that capture the kind of structured noise typical of cognitive and neural systems. Moreover, unlike the Bayesian models referenced above, the complex dynamical systems models of Haken and Ta'eed et al. do not require experimental manipulations in order to account for natural instances of bistable visual perception.

Despite the fact that the above provides reasons to utilize complex dynamical systems theory *instead* of Bayesianism, we think that the former can benefit from the latter as well. As we have argued, complex dynamical systems theory provides methods and theories that are closer to the nature of the systems investigated in the cognitive and neural sciences (e.g., nonlinearity) than is typical of Bayesianism (e.g., linearity). What Bayesianism does quite well, however, is provide methods for assessing the probabilistic characteristics of those systems. A complex dynamical systems theory model of bistable visual perception may account for the neurophysiology and (ecologically valid) experimental work quite well. However, what it does not do as well is provide a means to *predict* when a particular percept will be stable. For that reason, a *complex dynamical Bayesianism* can leverage the best of both worlds: develop models that are biologically realistic and ecologically valid, and have strong probabilistic predictive power.

## 5. Challenges and Future Directions

One of our main aims has been to demonstrate crucial shortcomings of Bayesianism, broadly construed, in the cognitive and neural sciences. Our other main aim has been to motivate an alternative approach that incorporates Bayesianism, while not suffering from those shortcomings: *complex dynamical Bayesianism*. Here, we put forward a number of challenges facing our approach and future directions. As with preexisting Bayesian approaches (Section 2), the nature of the relationship between complex dynamical Bayesianism and the cognitive and neural sciences would benefit from being precisified. First, what are the theoretical commitments of complex dynamical Bayesianism and do they differ from those underlying Bayesianism or complex dynamical systems alone? For example, it may be the case that complex dynamical Bayesianism lends itself more readily to non-brain-centric conceptions of cognition (e.g., distributed, embodied, and extended cognition) than Bayesianism typically has. Second, is the potentially complicated juggling act pertaining to how the statistics of complex dynamical Bayesianism ought to be interpreted. As discussed above, Bayesian statistics alone has various interpretations, one of which is frequentist. Complex dynamical systems theory commonly rejects the use of frequentist statistical assumptions. Does it then follow that complex dynamical Bayesianism cannot, in principle or practice, appeal to frequentist statistical assumptions? If yes or no, then why?

The third issue concerns the proper way to interpret complex dynamical Bayesianism modeling. Specifically, is it properly a *realist* or an *instrumentalist* investigative framework for the cognitive and neural sciences? If realist, then the issue arises as to how to justify treating the target of investigation as being both complex dynamical *and* Bayesian in nature. Doing so, however, raises issues such as those just mentioned above concerning statistical interpretations. For example, can a target of investigation being studied under complex

dynamical Bayesianism justifiably be modeled with seemingly incongruent analytical commitments? Related, can a multilevel phenomenon have differing characteristics at various scales (e.g., exhibit linearity at one temporal scale and nonlinearity at another) and still fall under the purview of complex dynamical Bayesianism? If yes or no, then why? Would such justification be attainable a priori or would such questions necessitate arbitration via empirical data? These questions are challenging and many are not specific to the framework currently on offer. Nonetheless, complex dynamical Bayesianism is unique with regard to such issues. If our assessment is correct that Bayesianism in the cognitive and neural sciences typically has a set of commitments that are incongruent with those of complex dynamical systems theory, then complex dynamical Bayesianism may be faced with the need to reconcile deep conceptual, methodological, and theoretical commitments that are irreconcilable. To conclude if that is the case or not will require deliberating upon both empirical and theoretical considerations.

### 6. Conclusions

Bayesianism and complex dynamical systems theory have both facilitated understanding of a multitude of cognitive and neural phenomena. As we have argued in this theoretical paper, while attempts have been made to incorporate nonlinearity and noise in standard Bayesian models, such work has been largely unsuccessful at maintaining the level of biological realism and ecological validity needed to properly account for cognitive and neural systems. The recommendations and illustrative examples from bistable visual percepts in the previous section provide a preliminary attempt at guiding the successful integration of Bayesianism and complex dynamical systems theory. Such guidelines can maintain core features of Bayesianism (e.g., predictive control, statistical estimation, and top-down processing) and capture a greater degree of biological realism via nonlinear dynamical concepts (e.g., historical variation, nonlinear patterns of behavior, and qualitative shifts). Understood in this way, a *complex dynamical Bayesianism* would facilitate additional research in the cognitive and neural sciences that explains more of the everyday natural world that follows the rule of nonlinearity and structured noise, and not the exception of linearity and random, unstructured noise (cf. [1]).

**Author Contributions:** Conceptualization, M.J.A. and L.H.F.; writing—original draft, L.H.F.; writing, M.J.A. and L.H.F.; writing—review and editing, M.J.A. and L.H.F. All authors have read and agreed to the published version of the manuscript.

**Funding:** This research received no external funding.

**Data Availability Statement:** Not applicable.

**Conflicts of Interest:** The authors declare no conflict of interest.

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
