# Peer review of "Enhancing Bayesian Approaches in the Cognitive and Neural Sciences via Complex Dynamical Systems Theory"

_2673-8716, doi:10.3390/dynamics3010008_

Round 1

Reviewer 1 Report

After reading the paper, I think the paper is well-written and organized. But I have some comments to be addressed before recommending the acceptance of the paper:

1. The abstract should be rewritten in an informative way to highlight the paper's main findings. 

2. Remove the lines before the introduction section, lines 26-32.

3. The organization of the paper should be mentioned at the end of section 1.

4. Section 2 is very long. The authors should try to divide it into subsections as possible.

5. It is known the bold symbols refer to matrices. Thus, Avoid using bold symbols as in equation 1.

6. I suggest adding the violin plot beside the normal curve in Fig 2.  

Reviewer 3 Report

The authors claim that Bayesianism has two interrelated shortcomings: its calculations and models are primarily linear, and noise is assumed to be random and unstructured. This article outlines how Bayesian theory addresses these shortcomings. First, the nonlinear features of biological cognitive systems are more concentrated. Second, instead of treating noise as random and unstructured dynamics, it is treated as structured nonlinearities of complex dynamical systems (e.g., chaos and fractals). If successfully integrated, the theory of complex dynamical systems will promote Bayesianism, which is more capable of explaining some natural phenomena that are currently out of reach, from a realistic perspective of biology. However, the description of the innovation leaves something to be desired.

The following suggestions could be of help to improve the paper’s quality.

Q1:Abstract

It is recommended that the author add a challenging description of the current work to make the work more innovative and necessary.

Q2:Introduction

It is hoped that the author can describe the challenge of the work in this paper in detail. In addition, it is suggested that the author focus on the innovation of the work in this paper.

Q3:Introduction

An increasingly-popular frontrunner in recent years is Bayesianism.

This view needs to be substantiated by relevant references

Q4:A Brief Introduction to Bayes’ Theorem

While not all applications of Bayesianism to cognitive and neural phenomena are constrained to linear models, those that are remain problematic for two major reasons

This view needs to be substantiated by relevant references

Q5:Conclusion

What are the other disadvantages of the enhanced Bayesian approach? What is the future direction of work?

Q6:The quality of English needs improving.

Round 2

Reviewer 2 Report

As I wrote, the concept of Bayesianism is understood very broadly, and I must say more broadly than I know it. The authors state that the concept is based on Bayes' formula, and this is part of probability theory based on measure theory on sets. I allow that the term Bayesianism may exist outside of it. Like me, there may be readers who understand the concept within probability theory, so that would need to be explained in the introduction.
